# Detecting Central Auditory Processing Disorders in Awake Mice

**DOI:** 10.3390/brainsci13111539

**Published:** 2023-10-31

**Authors:** Camille Dejean, Typhaine Dupont, Elisabeth Verpy, Noémi Gonçalves, Sabrina Coqueran, Nicolas Michalski, Sylvie Pucheu, Thomas Bourgeron, Boris Gourévitch

**Affiliations:** 1Institut Pasteur, Université Paris Cité, INSERM, Institut de l’Audition, Plasticity of Central Auditory Circuits, F-75012 Paris, France; 2Cilcare Company, F-34080 Montpellier, France; 3Sorbonne Université, Ecole Doctorale Complexité du Vivant, F-75005 Paris, France; 4Institut Pasteur, Université Paris Cité, CNRS, IUF, Human Genetics and Cognitive Functions, F-75015 Paris, France; 5CNRS, F-75016 Paris, France

**Keywords:** central auditory processing disorders, autism spectrum disorders, awake mice, auditory steady-state response

## Abstract

Mice are increasingly used as models of human-acquired neurological or neurodevelopmental conditions, such as autism, schizophrenia, and Alzheimer’s disease. All these conditions involve central auditory processing disorders, which have been little investigated despite their potential for providing interesting insights into the mechanisms behind such disorders. Alterations of the auditory steady-state response to 40 Hz click trains are associated with an imbalance between neuronal excitation and inhibition, a mechanism thought to be common to many neurological disorders. Here, we demonstrate the value of presenting click trains at various rates to mice with chronically implanted pins above the inferior colliculus and the auditory cortex for obtaining easy, reliable, and long-lasting access to subcortical and cortical complex auditory processing in awake mice. Using this protocol on a mutant mouse model of autism with a defect of the *Shank3* gene, we show that the neural response is impaired at high click rates (above 60 Hz) and that this impairment is visible subcortically—two results that cannot be obtained with classical protocols for cortical EEG recordings in response to stimulation at 40 Hz. These results demonstrate the value and necessity of a more complete investigation of central auditory processing disorders in mouse models of neurological or neurodevelopmental disorders.

## 1. Introduction

Functional tests for auditory dysfunction traditionally target the periphery because, in the vast majority of cases, losses of auditory sensitivity involve alterations to sound processing within the cochlea. For example, the distortion products of otoacoustic emissions reveal outer hair cell (OHC) dysfunction; tympanometry can be used to detect middle ear conductive hearing loss; auditory brainstem response (ABRs) recordings assess the normality of neural response amplitude and latency for each auditory nucleus between the auditory nerve and the inferior colliculus [1,2,3]. These methods have been widely used in both humans and animal models, mostly mammals, in which the physiology of the peripheral organs closely resembles that in humans.

However, recent years have seen a shift in focus from peripheral auditory disorders to auditory disorders with a possible central origin, i.e., from the cochlear nucleus in the brainstem to the auditory cortex (AC). In particular, issues with listening comprehension and poor performance in auditory processing tasks (such as understanding degraded speech) despite a normal audiogram broadly define what are referred to as *central auditory processing disorders* (CAPDs) [4]. Improvement in our understanding of CAPDs is required, given the high frequency of these disorders in elderly populations [5,6] and their presence in 2–5% of school-aged children [7], even though this diagnosis remains controversial [8]. CAPDs have also recently been associated with acquired neurological and neurodevelopmental conditions. The prevalence of CAPDs is high in patients with early Alzheimer’s disease or mild memory impairment [9,10], autism spectrum disorders (ASD) [11,12,13], and schizophrenia [14,15]. Mouse models reproducing human phenotypes now exist for all these conditions and many others [16,17,18,19,20], and new models appear every year. Large cohorts of mice can therefore be tested for CAPDs, potentially with follow-ups over time. However, we need to determine what to test, where to collect data, and under what conditions.

Currently, there is no “gold standard” test for CAPDs, and it is not clear whether these disorders really exist as independent entities separate from other neurodevelopmental disorders with overlapping symptoms [8,21]. However, periodicity coding, and auditory temporal processing more generally, are atypical or degraded in neurodevelopmental disorders [22,23] and acquired neurologic disorders [24,25]. In conditions such as autism, schizophrenia, and Alzheimer’s disease, an altered auditory steady-state response (ASSR) to specific stimulation at 40 Hz is thought to reveal dysfunctions of circuits supporting gamma activity and predict clinical outcome [26,27,28,29,30], even if this hypothesis remains to be confirmed in animal models. A degradation of the auditory temporal processing is also prevalent in elderly populations, and may contribute to a poor understanding of speech [31,32,33]. Dysfunctions in the brainstem or midbrain are thought to be overcompensated in the auditory cortex, in which enhanced responses are typically observed in both elderly humans and aging rodents [33,34,35]. The onset of the response is therefore of particular interest, together with auditory temporal processing, for the development of screening tests for CAPDs.

The collection of neural correlates of CAPDs in large cohorts of animals may also be challenging. The use of micro-electrodes involves heavy surgery that is both tricky and invasive and, therefore, not very compatible with the rapid screening and follow-up of large cohorts. Several noninvasive methods, such as middle-latency responses and cortical auditory evoked potentials (CAEPs), are already widely used in human studies to assess pharmacodynamics, cortical lesions, sensory processing, and clinical outcomes for auditory cortical function [11,36,37,38,39,40,41]. In rodents, particularly mice, in which CAEPs are recorded subcutaneously [42] or, more often, epidurally, only a few studies have measured middle- or long-latency CAEPs [43,44,45,46]. CAEPs are most frequently obtained using very simple stimuli such as pure tones, clicks, or noise bursts. However, like ABRs, CAEPs are typically obtained under anesthesia in animals, with a greater impact on central than peripheral processes [47]. This issue is particularly problematic for the measurement of CAPDs in genetic models of neurological or psychiatric diseases, as anesthesia disrupts top-down pathways [48] and neural oscillations [49], increases latencies [50], and depresses cortical responses [51,52,53], all of which may also be affected by the diseases under investigation. The mice, therefore, need to be awake during testing.

In this brief report, we present a provisional protocol for following ABRs, response onset, and temporal processing in both the inferior colliculus and auditory cortex over a period of weeks in awake mice, following a short surgical intervention. We used this method to investigate central auditory processing in mice lacking the major isoforms of SHANK3 (SH3 and multiple ankyrin repeat domains protein 3), a scaffolding protein of the glutamatergic synapse, whose deficit is associated with ASD in humans [54,55] and synaptic dysfunction and altered behaviors in mutant mice [56,57,58,59,60,61]. 

## 2. Materials and Methods

All aspects of this study were approved by the Institut Pasteur animal research ethics committee.

### 2.1. Animals

We used 17 *Shank3*^∆11/∆11^ (11 females, 6 males) and 30 *Shank3*^+/+^ animals (15 females, 15 males). *Shank3*^∆11/∆11^ mice were generated by Genoway (Lyon, France), with 129S1/SVImJ ES cells. The *Shank3* mutation, a deletion of exon 11, was then transferred into the C57BL/6J background, with introgression over more than 15 backcrosses [62]. In this model, certain minor isoforms of the protein are expressed [62]. The animals were 6 to 25 weeks old at surgery and weighed between 12 g and 34 g. The control and *Shank3*^∆11/∆11^ animals were of similar age (82 ± 35 d vs. 80 ± 40 d). In this age range, generally, peripheral auditory processing is yet to be affected by the C57BL/6J genetic background [63] or by the *Shank3* mutation (manuscript in preparation). The animals were placed in standard housing in 12-h light/dark cycle conditions, with a background noise of <40 dB sound pressure level (SPL). They had unlimited access to food and water and were housed in cages containing five mice each.

### 2.2. Surgery

Each animal underwent a 1.5-h surgical procedure with sterilized instruments. The animal was first sedated by subcutaneous injection of a mixture of medetomidine (0.1 mg/kg) and buprenorphine (0.1 mg/kg), and then anesthetized with isoflurane (4% for induction and then 2%, with 95% oxygen). The animal’s head was shaved, and an ophthalmic lubricant was applied to the eyes. Lidocaine (10 mg/kg) was injected subcutaneously directly above the skull. The skin was carefully cleaned with betadine and ethanol.

A clean incision was made on the skull, large enough for placement of three pins (two for recording and one for reference) and a holding headpost. The excess skin was removed and skin edges were glued to the skull with surgical glue. The vasodilator properties of lidocaine made it possible to observe the vessels under the skull and, thus, locate the areas of interest: the left AC and left inferior colliculus (IC). A hole was drilled above the AC with a compact milling machine (burr diameter 1.2–1.5 mm). A 0.6 mm pin (tin, 8 mm long, from a classical male pin header connector, RS Components, Corby, United Kingdom) was inserted into the hole in contact with the dura and was glued to the skull. The same method was applied for the pin above the IC and the reference pin was positioned above the right frontal area. Finally, the headpost was fixed to the skull with dental cement (Figure 1A).

### 2.3. Post-Surgery

We injected meloxicam (Metacam, 5 mg/kg, sc) before the animals woke up and then once daily (1 mg/mL) for three days. In case of infection, sulfadoxine-trimethoprim (Borgal, 20 mg/kg, sc) was injected. The weight of the animals had typically increased or stabilized by day 3 post-surgery. Buprenorphine injections were administered (1 mg/kg, sc), when necessary, after pain score evaluation. Animals were acclimated to head fixation for a few days before the experiments.

### 2.4. Electrophysiology Recordings

The animals were placed in a stereotaxic frame with the headpost fixed to the frame (Figure 1A) in an acoustically and electromagnetically isolated chamber. The epidural pins of the animal were connected to a TDT RA4 preamplifier and a Tucker–Davis Technologies (TDT) RZ6 recording station. The RZ6 also transmitted the sounds to a free-field (TDT ES1) transducer. Electrophysiological responses were recorded from epidural pins at a sampling rate of 24 kHz for ABRs and 5 kHz for EEG. Signals were subjected to a Butterworth filter between 3 Hz and 3 kHz and were displayed with TDT Biosigz (v5.7.4) software (ABRs) or Matlab 2022a (Mathworks). For ABRs, the sound stimuli were 5 ms pure tones with a 1 ms rising and falling ramp and frequencies between 5 kHz and 32 kHz. The rate of stimulation was 15 Hz, and each stimulus was typically presented 100–200 times. Only the pin located above the AC was used for ABRs. For EEG, spontaneous neural activity was recorded from the IC and AC for 5 min. We then presented 100 μs trains of 1 s clicks, at click rates between 2 Hz and 480 Hz, eight times for each rate, in a pseudorandom order and with an interstimulus interval of 2 s. Sounds were generated with Matlab (Mathworks).

For a given animal, the session lasted about 1.5 h, after which the animal was returned to its cage. Recordings were conducted three to four times for each animal, over a period of 10–40 days. For ABRs, all animals were suitable for use as they all had a threshold below 50 dB at 10 kHz. We therefore selected the session with the best average thresholds. For EEG, a few animals of both genotypes displayed no significant onset response to clicks (defined as an onset amplitude greater than four times the standard deviation of the spontaneous EEG response) in the AC or IC and were therefore excluded from the analysis. We recorded EEGs in the AC and/or IC in 12 *Shank3*^+/+^ animals and 12 *Shank3*^∆11/∆11^ animals. For each animal, we selected the best session (the session with the largest onset amplitude) from those recorded.

### 2.5. Statistical Tests

Estimation of neural response onset and the amplitude of the auditory steady-state response (ASSR) are described in the legend to Figure 1. The significance thresholds for ASSR (and for the other measurements of phase-locking used) were determined by calculating the 97.5th percentile of 1000 values simulated from pink noise with the same length and number of repetitions as the recorded data. Phase-locking was also measured by the cerebro-acoustic coherence (CAC) [64,65]. CAC measures the spectral coherence between the stimulus (the click train) and the neural response at the modulation rate of the click train. It is calculated for individual trials and is, therefore, less sensitive than other methods to the intertrial variability of signal shape, noise, and latency.

We analyzed the data by two-factor ANOVAs (genotype and frequency/SPL) and post-hoc *t*-tests. For all the tests reported here, readability was improved by inserting a reference (a small letter in subscript) linking to the full details of the test in Appendix B. 

## 3. Results

### 3.1. Monitoring Central Auditory Processing in Head-Fixed Awake Animals

We designed a protocol for testing central auditory processing with a limited number of stimuli and a simple analysis at two sites involved in different auditory stages (IC and AC) and in awake conditions. We imposed the additional constraints that the surgery had to be minimal and cheap, for compatibility with the testing of large cohorts of animals over long periods of time. We chose to implant standard electronic pins at the surface of the IC and AC, together with a homemade headpost, to make it possible to record from head-fixed animals in awake conditions (Figure 1A). This experimental design, with only pure tones and click trains as stimuli, can be used to measure ABRs (Figure 1B–D), auditory onset properties in the IC/AC (Figure 1C,D), and the phase-locking abilities of the IC/AC in awake mice by monitoring the auditory steady-state response to click trains (Figure 1C,E). Evoked responses can be monitored over periods of days or weeks, even if there are changes in the emotional state of the animal or in the experimental context (Appendix A).

**Figure 1 brainsci-13-01539-f001:**
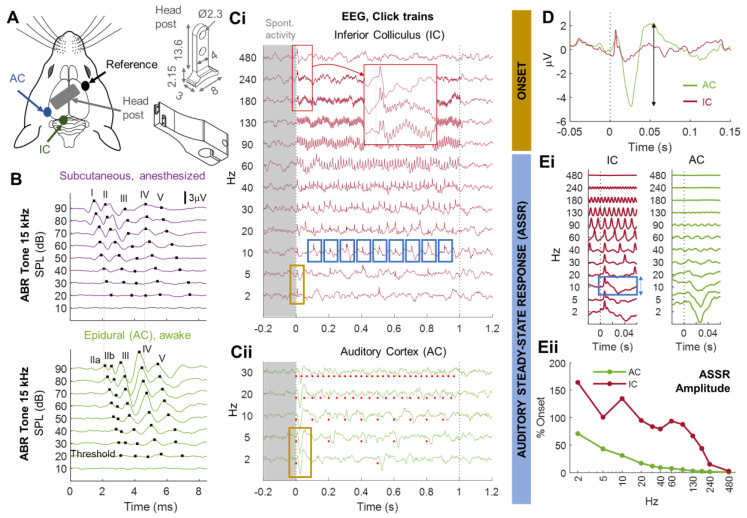
Overview of the method for monitoring central auditory processing in head-fixed awake animals. (**A**) Three pins were implanted epidurally through the skull above the left auditory cortex (AC), the left inferior colliculus (IC), and the right prefrontal cortex for reference (left scheme). A headpost (top right, dimensions in mm) was cemented on the surface of the skull and attached to the stereotaxic frame with a holding piece (bottom right). (**B**) Typical auditory brainstem recordings (ABRs) following stimulation with pure tones (here, 15 kHz). Waves II, III, IV, and V correspond to successive activations of the cochlear nucleus (CN), superior olivary complex (SOC), lateral lemniscus (LL), and IC, respectively. ABRs are typically recorded subcutaneously with an electrode located at the mastoid and in anesthetized animals in response to tones of different frequencies and different levels of intensity (purple, top). The lowest level of intensity eliciting a visible response was the auditory threshold for the tone frequency used (here, 20 dB SPL). The ABR thresholds of each animal were obtained at several frequencies across the animal’s hearing range. The same waves are visible on the AC pin implanted in (**A**) in awake conditions (green, bottom). Note that the midbrain waves (IV, V) are stronger than those measured subcutaneously at the mastoid. (**C**,**D**) Assessment of auditory onset and steady-state responses. (**C**) Click trains at rates between 2 Hz and 480 Hz elicited phase-locked responses up to hundreds of Hz in the IC (**C**)(**i**) and up to only 30–40 Hz in the AC (**C**)(**ii**). (**D**) Onset response to the first click averaged across repetitions of 2 Hz and 5 Hz click trains (yellow rectangles in (**C**)(**i**) and (**C**)(**ii**)). The responses of the IC and AC were tightly correlated for times < 10 ms. The range (black arrow) of the evoked response is defined as follows: (**E**)(**i**) averaged response to each click repetition (excluding the first one, see example for 10 Hz, blue squares in (**C**)(**i**)). This extended averaging provides an excellent signal-to-noise ratio for very high click rates and, therefore, a reliable estimate of the amplitude (blue arrow) of the auditory steady-state response (ASSR). (**E**)(**ii**) Amplitude of ASSR relative to the onset as a function of click rate. This function is also called the neural temporal modulation transfer function [66].

### 3.2. Central Auditory Processing of Shank3^∆11/∆11^ Animals

We used the protocol described in Figure 1 to investigate central auditory processing in a mouse model of autism: mice with a deletion of exon 11 of the *Shank3* gene.

The ABR thresholds of awake *Shank3*^∆11/∆11^ mice were slightly degraded^a^ (by about 10 dB up to 20 kHz^b,c,d^) relative to those of *Shank3*^+/+^ mice, but they remained very good (Figure 2A). We also plotted ABR wave latencies as a function of SPL for the tone frequency 10 kHz (best frequency in mice) and for all SPL for which we had a significant response in all animals, i.e., between 50 dB and 90 dB SPL (Figure 2B). As explained in the Figure 1 legend, wave I was not reliably identified in our epidural ABRs. Wave latencies (II, III, IV, V) were similar between control and *Shank*3^∆11/∆11^ animals^e^. 

We then recorded spontaneous EEGs for both the IC and AC. The EEG spectrum was qualitatively different between the two areas, with a greater proportion of energy in high-frequency bands for the AC rather than for the IC (Appendix A). No difference in energy between the two groups of animals^f^ was observed for any of the frequency bands, for either the AC or the IC (Appendix A). The spectral coherence of the IC and AC signals was also similar in the two groups of animals^g^, suggesting that there was no alteration of the transfer function between the IC and the AC (Figure 2C).

We then recorded the EEG in response to click trains at rates between 2 Hz and 480 Hz. The response to the first click, i.e., the onset range (peak minus trough), as calculated in Figure 1D did not differ significantly between control and *Shank3*^∆11/∆11^ animals in either area^h,i^ (Figure 2D(i,ii)). Onset latencies were also similar between the two groups of animals for both areas^j,k^ (Figure 2D(iii)).

However, ASSR amplitude (relative to the onset, as in Figure 1) was lower in the IC of *Shank3*^∆11/∆11^ animals than in the IC of control animals for click rates between 60 Hz and 130 Hz^l,m,n,o^ (Figure 2E(i),F(i)). The difference between the groups observed in the AC (Figure 2F(ii)) at such frequencies was not significant but was intriguing, particularly as ASSR amplitudes above 60 Hz barely emerged from the baseline amplitude for *Shank3*^∆11/∆11^ animals (gray area in Figure 2F(ii)). Indeed, the proportion of animals with a significant ASSR amplitude at 60 Hz was significantly higher for control animals than for *Shank3*^∆11/∆11^ animals^p^ (42% vs. 83%, “x” in Figure 2F(ii)). To confirm this result, we measured phase-locking in the AC with a very different method: cerebro-acoustic coherence (CAC) (see Section 2.5). Like the popular vector strength measurements applied to spike trains in studies relating to temporal modulation transfer functions [67] and the intertrial coherence widely adopted in studies of neurological disorders [68,69], CAC measurements are not reliable for short-duration recordings evoked by click trains at low rates. In such situations, the small number of cycles within 1 s of recording are overwhelmed by the large amount of neural “noise” at low frequencies, resulting in an artificial bandpass-shaped function for the CAC (Figure 2G(i)). Nevertheless, the measurement of CAC was pertinent in our context because our results for ASSR amplitude suggested deficits for rates above 10 Hz. As the IC can follow fast click rates in the 60–200 Hz range perfectly, the CAC almost reached 1 in this frequency range in this auditory region, whereas this was not the case in the AC (Figure 2G(i,ii)). Indeed, we found a clear trough in the click–rate–CAC function at about 60 Hz in the AC of *Shank3*^∆11/∆11^ animals that was not observed in control animals (Figure 2G(ii)). In *Shank3*^∆11/∆11^ animals, the CAC measurement at 60 Hz was significantly lower than those at 10, 20, 30, or 180 Hz^q^ (“o” in Figure 2G(ii)). In addition, the proportion of significant CAC measurements was lower in *Shank3*^∆11/∆11^ animals than in control animals only at 60 Hz^r^ (42% vs. 83%, “×” in Figure 2G(ii)). Finally, the CAC measurement at 20 Hz in the AC was higher in *Shank3*^∆11/∆11^ animals than in controls^s^. Note that these trends on ASSR and CAC remain true for the subset of youngest and oldest animals (Appendix A). These findings are also similar to those obtained using the classical intertrial coherence measurement, except that the intertrial coherence method is subject to sensitivity problems [70] and yields higher significance thresholds with simulated data than the CAC method (compare Figure 2G and Appendix A). Overall, these results suggest that the neural response to click trains between 60 Hz and 130 Hz is lower in the IC, but there is a strong deficit in phase-locking to the specific frequency of 60 Hz in the AC in a large proportion of *Shank3*^∆11/∆11^ animals.

## 4. Discussion

### 4.1. Technical Advantages and Limitations

In mice, as opposed to other rodents widely used in the laboratory, such as guinea pigs, gerbils, and rats, the IC is located at the surface of the brain and is, therefore, accessible to epidural pin electrodes. This makes it possible to record populational signals, such as the ECoG signal, simultaneously in the IC and AC—two areas separated by two synapses—with minimal surgery and in awake animals. The protocol described in Figure 1 can be used for the rapid screening of certain common CAPDs, such as degraded temporal processing (via the ASSR) [22,23,24,25,31,32,33], 40 Hz-related dysfunctions [26,27,28,29], and changes in central gain (via the comparison of onset between the IC and AC) [33,34,35]. Beyond developmental abnormalities, CAPDs also occur in various conditions, such as tumors, strokes, head trauma, epilepsy, and metabolic disorders [10]. We also recorded spontaneous EEG signals because abnormal *spontaneous* oscillations have been observed in the AC of mouse models of neuropsychiatric conditions, such as fragile X syndrome [71] and schizophrenia [72,73]. The epidural pins also made it possible to record ABRs in awake animals, which can be used as a control for normal peripheral auditory function up to the IC, as shown here in *Shank3* mutant mice.

In epidural ABRs, we labeled the early waves as IIa/b, III, IV, and V because their latencies corresponded roughly to those of the corresponding waves of classical subcutaneous ABRs in mice (see Figure 1C and [74,75]). We reasoned that with a subcutaneous electrode at the mastoid, as in ABRs, there might be a conduction time for late waves, such as waves IV and V, which would therefore probably occur slightly earlier if recorded locally in the brainstem/midbrain. This is, indeed, what we observed in our epidural recordings, in which the large late waves occurred slightly earlier than in subcutaneous recordings (and under anesthesia). However, it remains to be confirmed whether this correspondence is real; alternatively, our wave IIa may actually be a delayed wave I.

We were surprised to observe ABR-related waves in AC epidural signals, as, to our knowledge, this phenomenon has never been reported or used in studies of epidural CAEPs. It is likely that early ABR waves are invisible for the typical sampling rate < 1 kHz and low-pass filtering < 300 Hz used in EEG recordings. One study recorded the local field potential with microelectrodes in the IC and observed the presence of the early waves (I–V) only with wide-band (30–3000 Hz) filtering [74]. Stimuli must also be presented many times, as illustrated by the absence of visible ABRs in the click-evoked onset response in the AC in Figure 1D, which was obtained with only 16 repetitions (using 2 Hz and 5 Hz click trains, see Figure 1D legend). If our putative wave II is not a delayed wave I, as suggested above, then wave I of classical subcutaneous ABRs, accounting for auditory nerve activity, is not reliably observed in our epidural ABRs. However, the putative waves III, IV, and V of epidural ABRs have a larger amplitude than those obtained with a subcutaneous mastoid electrode (Figure 1B).

We considered the possibility that the EEG response in the AC might be contaminated by activity from the IC, as electrical conduction could potentially occur between the two implanted pins, which are located close to each other. However, we are confident that such contamination is negligible because the response to click rates > 30 Hz disappeared for the AC pin but not for the IC pin. Conversely, one obvious advantage of our system is that the fixed location of the electrodes reduces the positional bias observed for subcutaneous recordings made over periods of days, increasing the reliability of longitudinal studies. Provided that bone growth or inflammatory processes in the animal do not intrinsically degrade the contact between the pin and the brain surface, we are confident that these pin implantations will prove very useful for the long-term follow-up of animals with progressive CAPD as is the case for those occurring during aging. Mismatch negativity or oddball paradigms could also be included in our battery of stimuli, as they are also frequently used for the diagnosis of CAPD during childhood [7].

Finally, we felt that it was important to test a wide range of click rates as, in many studies, ASSR assessments are limited to the 20–50 Hz range [72,76,77,78]. We found that genotype had an effect only on phase-locking to click rates at and above 60 Hz, with no effect observed at 40 Hz, the click rate typically used for testing. 

### 4.2. Shank3^∆11/∆11^ Mice as a Model of ASD

We used our protocol design to study CAPD in an awake mouse model of ASD. ASD is a heterogeneous pervasive developmental disorder of high prevalence (~14.7/1000 in children under the age of eight years). It is characterized by abnormal repetitive and stereotypic behaviors, restricted interests in early childhood, and a deficit in social communication [11]. Most children with ASD have some degree of sensory dysfunction [79].

SHANK3 is a scaffolding protein located in the postsynaptic density of excitatory synapses and crucial for synapse development and plasticity [80]. *SHANK3* mutations cause many neuropsychiatric disorders, including the Phelan–McDermid syndrome (which arises from a deletion or point mutation in one copy of the *SHANK3* gene), ASD, and schizophrenia [81]. *SHANK3* mutations are detected in 1–2% of patients with autism and intellectual disability, identifying the deletion or inactivation of one copy of this gene as a major cause of neurodevelopmental disorders [54]. Epilepsy is also a common comorbidity [81,82]. However, the mechanisms underlying these symptoms remain largely unknown. Interestingly, children with the Phelan–McDermid syndrome have a poorer neural response to communicative vocal sounds than children with idiopathic ASD [83]. This finding suggests that *SHANK3* could be targeted in screening for deficits in neural representations of acoustic cues related to speech, such as the temporal envelope.

Mice with SHANK3 deficiency display various behavioral deficits such as enhanced self-grooming, novelty-induced anxiety, and learning and memory deficiencies [16,58,59,60,61,84,85]. These issues are typically more pronounced in homozygotes than in heterozygotes [80,84]. In our genetic model (deletion of exon 11 of *Shank3*), mice have also been shown to display repetitive and stereotyped behaviors, impaired social activity, and altered vocalization burst structures [56,57,61,80]. 

### 4.3. Spontaneous and Evoked EEG Signals in the Auditory Pathways of Shank3^∆11/∆11^ Mice

The neural correlates of these behavioral alterations have been little explored. Synaptic dysfunction seems to be consistently observed: *Shank3* mutant mice have low basal levels of glutamatergic neurotransmission in the hippocampus, altered striatal postsynaptic function, and impaired long-term potentiation (LTP), but not long-term depression (LTD) in corticostriatal circuits [58,60]. However, neural response studies in *Shank3* mutant mice remain scarce. In humans, *SHANK3* mutations increase the likelihood of epilepsy and of atypical absence seizures in particular [86,87]. ASD has been studied in greater detail and consistent functional disorders have been described in humans: low ASSR at 40 Hz [29] and in the gamma band [88,89] (but see [90]) in the primary AC of children, anomalies of high-frequency gamma-band oscillations (30–80 Hz) in the core auditory areas [91], delayed auditory evoked potentials (in response to pure tones) for the N100 wave [92] but not for P300 [93]. In mouse models of ASD, a delayed auditory evoked potential and a lower coherence between trials (i.e., a decrease in evoked response reliability) have been consistently reported in the gamma band [88].

At odds with published results, many of the measurements investigated here did not distinguish between groups of animals. *Shank3*^∆11/∆11^ mice did not reproduce the phenotype of prolonged ABR wave V latency found in infants and children with autism [94]. The putative wave V in our ABR data did not differ between mutant and control mice (Figure 2B), and the onset response latency in the IC, the putative source of wave V [74], was also similar across genotypes (Figure 1D). We also found no delayed or prolonged cortical onset responses in *Shank3*^∆11/∆11^ animals. The onset response was also of similar amplitude between groups, for both the IC and AC (Figure 2E(i)). The resting EEG did not show the increase in gamma power reported in some other ASD mouse studies [95], including studies on *Shank3b* models [96]. The coherence between the IC and AC, which measures a linear subcortical contribution to AC field potential, was also similar between groups. There have been reports of low levels of (spontaneous) gamma band activity in the AC in ASD, but we observed no such deficit in either the AC or IC (Appendix A). However, these results are not altogether surprising. Indeed, for the auditory function only baseline auditory startle responses have been found to be impaired in various *Shank3* exon deletion models [97]. Despite conflicting evidence for somatosensory function, other senses, such as vision, olfaction, and vestibular function, have been reported to be unaffected in rodent models based on *Shank3* variants [97]. 

### 4.4. ASSR in Shank3^∆11/∆11^ Animals

However, we observed one striking difference between control and mutant mice: the neural response to click rates between 60 Hz and 130 Hz was degraded in both the IC and the AC of *Shank3*^∆11/∆11^ mice relative to control mice. More precisely, we found that the strength of the evoked response (Figure 2F(i)) to such clicks was lower in the IC and that, possibly due to this, there was a phase-locking deficit at 60 Hz in the AC (Figure 2F(ii),G(ii)). To our knowledge, only one study has examined the neural representation of periodic sounds in the AC of rats heterozygous for a *Shank3* mutation [98]. The authors reported lower discharge rates in the neurons of several tonotopic AC areas in response to repeated noise bursts presented at equivalent rates of 7–13 Hz. Indeed, spontaneous activity and evoked responses to tones or speech sounds were typically weaker in mutant rats, despite the normal auditory thresholds of these animals. The authors suggested that these findings might reflect difficulties experienced by the rats when dealing with rapid speech stream processing or perception, but they did not test faster presentation rates. Our results point in a different direction that is more consistent with observations in human autism studies. Indeed, in humans, a dysregulation of neural oscillations, particularly for high-frequency gamma-band oscillations (30–80 Hz), has also been reported across the visual, auditory, and somatosensory domains in ASD [29]. Specifically, in response to click trains, at least three studies on ASD and one case study on a patient with a *SHANK3* defect reported a decrease in ASSR at around 40 Hz [89,99,100,101], at least in late childhood [102].

The leading hypothesis concerning the underlying mechanisms would be that changes in activity at 60 Hz reflect an atypical excitation–inhibition balance. Indeed, gamma-band activity depends on synchronized interactions between GABAergic parvalbumin-positive interneurons and pyramidal cells via the NMDA receptors of the interneurons [103,104,105,106]. However, SHANK proteins are present principally at excitatory synapses, rather than inhibitory ones, and SHANK3 deficiency decreases the neuronal excitability of pyramidal neurons [97]. Our results, therefore, likely reflect defaults in circuitries composed of excitatory pyramidal cells and inhibitory parvalbumin-positive basket cells (pyramidal-interneuron network gamma model, PING, [107,108]) rather than reciprocal inhibitory interactions between inhibitory cells. In contrast with the deficit at 60 Hz, we observed a better phase-locking at 20 Hz, only in the AC, at least with CAC, Intertrial Coherence and Vector Strength measures (see Figure 2G and Appendix A). It is, therefore, possible that neural circuits with time constants compatible with 20 Hz firing are more active than in control animals, for instance those involving somatostatin expressing interneurons [109]. Since the beta band range of oscillations at 20 Hz is rarely observed in the AC [110], these circuits would act at a global rather than local scale.

ASSR degradation above 60 Hz already occurs in the IC, which suggests that subcortical alterations may also be at work. Several results support this hypothesis. First, *Shank3* is expressed in the mouse midbrain, albeit not strongly [111]. Second, molecular studies have shown that there may be changes in the volume of auditory subcortical structures or the density of the neurons within them in various genetically or environmentally triggered rodent models of ASD [95]. In particular, SHANK3-defective rats have lower neocortex and IC volumes [112]. Third, even if this is, to our knowledge, the only ASD model tested, the neural responses in the IC of *Cacnα2δ3*^−/−^ mice followed fast sound presentation (at rates around 100 Hz) poorly [113]. However, there is not yet a consensus concerning functional alterations to the subcortex in humans with ASD and rodent models of this condition. Measurements in children with autism have revealed a trend towards lower amplitudes and longer latencies for late ABR waves [95]. By contrast, most studies on rodent models of autism found no change in ABR P1 waves or latencies, except for late waves in *Cntnap2* Fragile X syndrome models, and, even then, only in young individuals [95]. ABRs in rodents are recorded under anesthesia, a state that may dramatically decrease and delay ABR waves [114] (and CAEPs [115,116]). We believe that the recording of ABRs and CAEPs/EEG in awake animals is a necessary breakthrough for functional investigations of central auditory processing in rodents. 

The alterations observed in the IC in *Shank3* mutant mice also raise questions about the source of changes to the cortex. Indeed, the IC is a mandatory auditory relay that extracts and integrates temporal features of auditory signals from the periphery and auditory brainstem nuclei before projecting onto the auditory thalamus, which, in turn, projects onto the core AC. For this reason, AC responses are highly dependent on the responses of the IC. It seems likely that many studies have misinterpreted results observed in the cortex as originating from this region when similar alterations had already occurred but were not recorded at the subcortical level. Indeed, any alteration in the IC is likely to be either reflected or compensated in the AC. Our results suggest that further investigation into the synaptic properties of the IC and possibly also into the brainstem of *Shank3* mutant mice is required, ideally on a complete knock-out model [84,117].

## 5. Conclusions

We designed a protocol for assessing and monitoring changes in central auditory processing in awake animals. In doing so, we recorded the EEG response to various click rates at the subcortical and cortical levels in an awake mouse model of autism (*Shank3*^∆11/∆11^) over weeks. We found that phase-locking at high sound repetition rates (above 60 Hz) was degraded in the AC, but that this alteration was already visible in the IC. These results suggest that alterations of the post-synaptic density of glutamatergic synapses in *Shank3* mutant mice may affect fast temporal processing and at more auditory stages than previously thought. 

## Figures and Tables

**Figure 2 brainsci-13-01539-f002:**
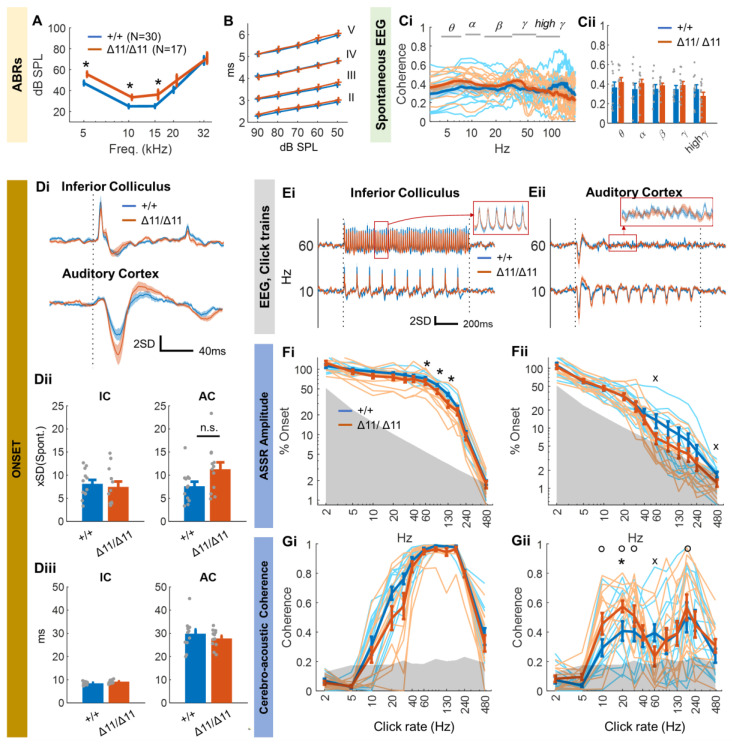
Central auditory processing in *Shank3*^∆11/∆11^ animals. (**A**) ABR thresholds and (**B**) ABR wave II-V latencies as a function of SPL for control (blue) and *Shank*3^∆11/∆11^ (orange) animals. *: *t*-test *p* value < 0.05. (**C**)(**i**) Coherence between spontaneous EEG in the left inferior colliculus (IC) and the left auditory cortex (AC). Thick lines indicate the mean values, whereas thin lines are individual curves. (**C**)(**ii**) Mean coherence per frequency band. (**D**) Onset response to the first click. (**D**)(**i**) Time response averaged over all animals. We normalized data between animals before averaging, by dividing the time signal for a given animal by the standard deviation (SD) from the spontaneous EEG taken for the 500 ms preceding the stimulus. The units for the *y* axis are, therefore, SD. (**D**)(**ii**) Amplitude range of the onset response. (**D**)(**iii**) Latency of the peak onset response. (**E**) EEG evoked by click trains at 60 Hz and 10 Hz and averaged over all animals in (**E**)(**i**) the IC and (**E**)(**ii**) the AC. (**F**) ASSR amplitude in (**F**)(**i**) the IC and (**F**)(**ii**) the AC relative to the onset response. (**G**) same as (**F**) for cerebroacoustic coherence (CAC). (**F**,**G**) Thick lines indicate the mean values and thin lines are individual curves. The gray area corresponds to the 97.5th percentile of the ASSR calculated for simulated data (see Section 2.5). “*”: *t*-test control vs. mutant animals *p* value < 0.05. “x”: proportion test for significant measurements, *p* value < 0.05. “o”: *t*-test for a given rate vs. 60 Hz for mutant animals, *p* value < 0.05. (**A**–**F**) The error bar or the shaded area around the mean indicates the standard error of the mean.

## Data Availability

The datasets generated for this study are available, on request, from the corresponding author.

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
