# Peer review of "Detecting Central Auditory Processing Disorders in Awake Mice"

_brainsci, 2023, doi:10.3390/brainsci13111539_

Round 1

Reviewer 1 Report

Comments and Suggestions for Authors

The authors present an original protocol for recording and monitoring central auditory evoked potentials in awake mice. It is a well-structured, valuable article, with a detailed presentation of the method and recorded data. The introduction presents the usefulness of CAPDs and the difficulties of recording, hence the need to find an easier way to record CAPDs in awake mice with Shank3 specific mutation for ASDs.

From my point of view, there would be only a few minor revisions to do

-Row 46 "understanding degraded speech" should be rephrased

-Row 146 - I think we cannot talk about an audiogram in animals but about ABR thresholds

-Row 242 - I think you should mention the "cerebro-acoustic coherence (CAC)" method in the Materials and Methods chapter

Comments on the Quality of English Language

Maybe some minor editing of English language required

Author Response

The authors present an original protocol for recording and monitoring central auditory evoked potentials in awake mice. It is a well-structured, valuable article, with a detailed presentation of the method and recorded data. The introduction presents the usefulness of CAPDs and the difficulties of recording, hence the need to find an easier way to record CAPDs in awake mice with Shank3 specific mutation for ASDs.

From my point of view, there would be only a few minor revisions to do

-Row 46 "understanding degraded speech" should be rephrased

We rephrased as “issues with listening comprehension”

-Row 146 - I think we cannot talk about an audiogram in animals but about ABR thresholds

An audiogram is implicitly a behavioral audiogram and has therefore been used for animals too for which behavioral thresholds can be measured (see works by Heffner mainly). The term is often used in a broader sense to describe auditory thresholds even in animals whatever the method used. Nevertheless, we agree that using the term “ABR thresholds” makes more sense here as we described ABRs above. We replaced audiogram by ABR thresholds as requested.

-Row 242 - I think you should mention the "cerebro-acoustic coherence (CAC)" method in the Materials and Methods chapter

In agreement with the reviewer, we moved the first description of CAC to the Materials and Methods chapter.

Comments on the Quality of English Language

Maybe some minor editing of English language required

A native English speaker checked all the manuscript and made a few changes.

Reviewer 2 Report

Comments and Suggestions for Authors

Dear authors,

Thanks for the submission. This research designed a protocol for assessing and monitoring changes in central auditory processing in awake animals targeting ABRs and CAEPs/EEG which might be a functional investigation breakthrough in other neurobiological disorders as well.  Meanwhile also compare the Shank 3 DKO with wild-type mice with the current methodology to study the central auditory processing in awake mice. However, some questions still need to be further addressed.

1: The Shank 3 DKO mice have a broad age range (80±40), is there any preliminary data that shows these different age mice showed a similar phenotype? Please provide more details in the 2.1 Animals section.

2: In Fig.1B, wave III which corresponds to superior olivary complex (SOC) is stronger in awake conditions as well, not just the midbrain waves (IV, V) are stronger than those measured subcutaneously at the mastoid. Any explanations for it?

3:  In Fig. 2B, any reasons why the wave II delayed latency is the only variant (wave III, IV, and V remain unchanged) in Shank 3 DKO compared to the control animals?

4: In Figure 2Gii, any hypothesis for why the Shank 3 DKO CAC measurement was lower than control only at 60Hz and higher at 20Hz? 

Author Response

Thanks for the submission. This research designed a protocol for assessing and monitoring changes in central auditory processing in awake animals targeting ABRs and CAEPs/EEG which might be a functional investigation breakthrough in other neurobiological disorders as well.  Meanwhile also compare the Shank 3 DKO with wild-type mice with the current methodology to study the central auditory processing in awake mice. However, some questions still need to be further addressed.

1: The Shank 3 DKO mice have a broad age range (80±40), is there any preliminary data that shows these different age mice showed a similar phenotype? Please provide more details in the 2.1 Animals section.

We used sexually mature young animals in which the peripheral auditory processing is hardly affected yet by the C57BL/6J genetic background or by the Shank3 mutation. Indeed, the C57BL/6J mice develop a late onset progressive hearing loss, due to a sensory hair cell bundle defect (Keithley et al, 2004), and the Shank3∆11 mutation also has an effect, only in aged animals, on the ABR thresholds in the high frequencies (E. Verpy, manuscript in preparation). We added these remarks in the Animals section. The difference of ABR thresholds between the two strains is less than 10dB in our study (Fig. 2A) and is therefore not responsible for the degraded temporal processing at suprathreshold level observed in Shank3 mice. Nevertheless, to confirm that our results do not depend on animal’s age, we provided a new supplementary figure (S3) showing results of Fig. 2FG for animals of age lower and greater than 90 days, resulting in the same trends as observed on Fig. 2FG.

2: In Fig.1B, wave III which corresponds to superior olivary complex (SOC) is stronger in awake conditions as well, not just the midbrain waves (IV, V) are stronger than those measured subcutaneously at the mastoid. Any explanations for it?

We thank the reviewer for this interesting point. First, we are confident that our so-called wave III was indeed associated with the SOC because the third ABR wave typically is even in other mammals (Melcher and Kiang, 1996) and because its latency (3-3.5ms) was consistent with those observed in other mouse studies (Henry, 1979; Aedo et al, 2016; Shearer dissertation, 2012; Toyoshima et al, 1009; Liu et al, 2019 etc..). The amplitude of a recorded wave depends not only on the proximity to the source generator but also on the propagation medium properties between the source and the electrode and the orientation of sink-source dipoles generating the voltage deflection. We don’t have an explanation on the much larger trough following wave III positive peak in awake animals but in addition to the spatial proximity of the electrode to the SOC, we suppose that the circuits generating this SOC deflection are aligned in a way such that the electrode placed at the surface of the AC “sees” a large voltage variation. For instance cortical columns of the neocortex are aligned so that the local field potential is large but such columns are absent in the IC where the local field potential is typically much smaller than in the cortex. To our knowledge, we are the first to record awake ABRs using pins in the AC and therefore there has been no investigation on this phenomenon previously. We think that given the absence of solid explanation, it is likely wiser to leave that phenomenon to further investigations rather than to consider a lengthy discussion in the manuscript.

3:  In Fig. 2B, any reasons why the wave II delayed latency is the only variant (wave III, IV, and V remain unchanged) in Shank 3 DKO compared to the control animals?

We were also puzzled by this result and went back on individual data after the reviewer’s comment. We found out that wave I (which is usually invisible in our awake ABRs) was actually visible on a few control animals and that wave I and II had been sometimes mislabelled on these animals. We thus decided to reanalyze all the ABR data of our animals, leading to an update of Fig. 2B on which the initial difference for wave II between groups of animals vanished and is not significant anymore. We are sorry for this mistake and thank the reviewer for pointing this anomaly to us.

 4: In Figure 2Gii, any hypothesis for why the Shank 3 DKO CAC measurement was lower than control only at 60Hz and higher at 20Hz?

We lengthily discussed the case of 60Hz (“Indeed, in humans… also be at work.”). The effect at 20Hz is replicated by ITC and VS measures but not by the ASSR (Supp. Fig. S4) and only in the AC. A good phase locking at 20Hz would mean that neural circuits having time constants compatible with 20Hz are more active than in control animals, for instance those involving somatostatin expressing interneurons (Chen et al, 2017). One difficulty is to decipher which circuits are involved in phase locking. With regards to oscillations, 20Hz is in the beta band, which typically emerges from active thinking (such as conditioning) as well as static motor control (Tan et al., 2016). It is mostly found in the motor cortex, the hippocampus, the prefrontal cortex or olfactory pathways but is rare or even absent in the auditory system (Gourevitch et al, 2020). One can therefore speculate that a better phase locking in the auditory cortex involves more global circuits than local ones. We added this to the discussion.

Round 2

Reviewer 2 Report

Comments and Suggestions for Authors

Dear authors,

Thank you so much for the detailed explanations. Some puzzles still need to be investigated which is better to be addressed in further research. For now, I don't have any other questions for this manuscript.

Thank you